# OpenReview forum: "Utilizing LLM Robustness for LVLM Safety via Reducing the Pretraining Modality Gap"
_ICLR.cc/2026/Conference — Submitted to ICLR 2026_

### Official Review · Reviewer_1y89 · 2025-10-25

**Soundness:** 2
**Presentation:** 3
**Contribution:** 2
**Rating:** 4
**Confidence:** 5

**Summary:**

The paper investigates the link between the modality gap in VLMs and safety degradation. It shows that a larger modality gap correlates with lower safety and that this gap mainly originates during pretraining. To address this, the authors propose REGAP, a pretraining regularizer that minimizes the distance between image and text token embeddings to reduce the modality gap. REGAP improves safety across multiple benchmarks while maintaining similar utility to the baseline. The method generalizes across architectures and can be combined with existing defenses for additional gains.

**Strengths:**

- The paper is well-presented and logically structured, offering an enjoyable and smooth reading experience.

- The idea of closing the modality gap without requiring any safety data, unlike most post-training safety methods, is interesting and represents a potential step toward inherent mitigation of cross-modality safety misalignment by pretraining VLMs this way from scratch.

- The experimentation across multiple models and a wide range of safety benchmarks, combined with ablation studies examining layer placement, scaling factors, and interactions with other defenses, is comprehensive and commendable.

**Weaknesses:**

Despite the strengths and comprehensive experiments, I have several serious concerns regarding the proposed method and its claims.

- The authors repeatedly claim that existing safety post-training techniques are computationally expensive, yet their own approach requires VLM pretraining (i.e., retraining the projector with the regularization loss), which is equally if not more costly. The efficiency claims, therefore, seem questionable, as several post-training methods, such as safety fine-tuning or unlearning, also freeze the main architecture and train only a small LoRA adapter, making their cost comparable.

- The proposed method requires redoing the pretraining stage, which is somewhat unrealistic since VLMs are typically already pretrained, and then fine-tuned and publicly available. In such cases, post-training interventions are far more practical, as it is generally infeasible to repeat the entire pretraining process for each model to ensure safety. Moreover, the authors themselves show that their method does not work when applied to fine-tuned models and is effective only during pretraining, further limiting its applicability as a post-training technique.

- The authors show that the modality gap between text and image embeddings is negatively correlated with model safety. However, this finding is heavily emphasized throughout the paper and investigated a lot, despite already being established in prior work, particularly the "CoCA: Regaining Safety-awareness of Multimodal Large Language Models with Constitutional Calibration" paper, which demonstrated the same relationship.

- The effect of the proposed training on utility preservation and other model capabilities remains unclear. The paper quantitatively evaluates utility only for LLaVA-7B-LoRA in Table 1. For other models (MiniGPT-4 and ShareGPT4V), the claim of “preserved utility” is stated but not empirically supported with benchmark results or numerical comparisons, as these are missing from Table 2. It would be helpful to clarify the reason for this omission. Moreover, reporting benchmark-wise utility scores rather than only aggregate values would make the impact of the proposed training method on different capabilities more transparent.

- According to Figure 2(f), the modality gap primarily exists up to layer 6 before applying REGAP. Could you provide an explanation for this behavior? How does it affect model safety, and what insights can be drawn from it? Is this the main reason you decided to minimize the modality gap only at the first layer? If so, why not extend it to multiple early layers? In Section 7, the ablation studies mention that regularizing the later layers worsens safety, but this observation is not clearly explained. It would be valuable to explore and justify why this occurs. Additionally, why does this regularization work only during the pretraining stage, yet corrupts the model when applied at the fine-tuning stage? These aspects remain unclear and would benefit from deeper investigation and discussion.

- It would be helpful to report and visualize the MIR on additional datasets, such as FigStep, after training to verify whether the embeddings of images containing harmful text and those of the corresponding textual prompts are truly aligned, and whether the modality gap has been fully closed.

- There is a minor issue with the template, which refers to ICLR 2025 instead of ICLR 2026.
Additionally, a few typos appear throughout the paper, such as:
– “LLaVA1.57B, QWen2.5VL” -> (LLaVA1.5-7B, Qwen2.5-VL)
– “and the unsafe rate for different LVLM after” -> (LVLMs)

- Given the adversarial nature of this work and in line with ICLR’s guidelines, it would have been preferable to include a dedicated Ethics Statement as well as the Reproducibility section in the paper.

**Questions:**

Please refer to the weaknesses.

---

> ### Author Response · Authors · 2025-11-25
>
> We thank the reviewer for acknowledging the clarity of our presentation, the novelty of addressing modality-gap reduction without safety data, and the comprehensiveness of our experiments across models, benchmarks, and ablation settings.
>
> * 1 & 2: ReGAP requires re-doing the VLM pretraining, and thus is more expensive than post-training methods.
>   * Unlike safety fine-tuning methods that require an additional training stage and safety data, ReGAP is designed to "replace" the standard LVLM pretraining, not to add an additional stage on top of it. Instead of starting from a pretrained model, we pretrain (with ReGAP regularizer) from an uninitialized model on the original pretraining data, and this regularized pretraining adds only \~5% additional training cost (Table 1). In this setting, “lightweight” refers to the pretraining cost relative to the original LVLM pretraining pipeline (i.e., 1.05). Compared to other methods that require substantial extra training on top of existing models, such as RobustCLIP, our approach is significantly more efficient. As we have shown, robust vision encoders or inference-time methods (which require safety data and some processing to e.g. find a steering vector or purify the model’s inputs or outputs) can be easily stacked with ReGAP to further boost safety.
> * 3\. The key observation (modality gap correlates with safety) is already known from prior work like CoCA.
>   * We have already discussed (see line 199\) that prior work, including CoCA \[1\] and CMRM \[2\] have contributed VLM safety degradation to distribution shift between text and text+image embeddings. However, our work shows for the first time that (1) the “amount” of modality gap is strongly correlated with “unsafe rate”. Furthermore, we demonstrated that (2) the modality gap is introduced during pretraining and (3) reducing the pretraining modality gap directly improves downstream VLM safety after fine-tuning, providing stronger evidence of a causal link rather than merely restating prior intuition. The above observations are made in our paper for the first time and were not established by prior work, including \[1\] and \[2\].
> * 4\. The paper shows utility metrics only for LLaVA, and benchmark-wise utility scores are not reported.
>   * As mentioned in lines 243–244, the initial MiniGPT4 and ShareGPT4V experiments used only a subset of their training datasets, so we did not evaluate their full utility scores at that stage.
>
> | Model | Unsafe Rate (↓) | Utility (↑) |  |  |  |  |  |  |  |
> | ----- | :---: | :---: | :---: | :---: | :---: | :---: | :---: | :---: | :---: |
> |  | HADES Orig | MMB | MMStar | QBench | GQA | SEED | SQA | Hall | Avg |
> | MiniGPT4 | 68 | 49.5 | 30.7 | 41.7 | 27.9 | 38.6 | 51.8 | 24.1 | 37.8 |
> | ReGAP | 59.8 | 49.3 | 28.7 | 40.5 | 24.2 | 40.3 | 55 | 29.8 | 38.3 |
>
>   * We trained MiniGPT4  on its full training data. Notably, MiniGPT4  shows a decreased unsafe rate with slightly better utility score, supporting the generality of our method. In addition, as noted in line 364, the full benchmark-wise utility scores for all models are reported in Table 11 in the appendix in our original submission. As shown, ReGAP consistently preserves utility, demonstrating our method does not degrade overall performance.
>   * For ShareGPT4, full-dataset training requires roughly 384 GPU-hours with NVIDA A40. We are still in the process of completing all updated runs. We will report the results if we get them in time, otherwise we will include the complete results in our final version.
>
> * 5\. The modality gap primarily exists up to layer 6 before applying ReGAP, and the reasons, implications for safety, and design choices are unclear.
>   * First, we note that the modality gap still exists in deeper layers, although it becomes much smaller and thus is not clearly visible in Fig 2(f). The full MIR of all layers of the LLaVA model after pretraining (provided in the revision, Table 8 and 9\) shows that the gap remains significant until the middle layers (although still larger than zero at final layer).
>
> | \# Layer | 1 | 4 | 7 | 10 | 13 | 16 | 19 | 22 | 25 | 28 | 31 |
> | :---- | ----: | ----: | ----: | ----: | ----: | ----: | ----: | ----: | ----: | ----: | ----: |
> | LLaVA-PT | 1080.27 | 136.36 | 35.44 | 13.93 | 6.96 | 3.6 | 2.1 | 1.57 | 1.33 | 1.27 | 1.23 |
> | LLaVA-FT | 304.44 | 37.06 | 10.09 | 5.65 | 3.82 | 2.55 | 1.84 | 1.51 | 1.37 | 1.28 | 1.21 |
>
>   * This aligns with the behavior described in the original MIR paper: as layers deepen, LVLMs progressively align visual and textual distributions to incorporate the image modality into the language stream. Shallow layers receive raw image representations from the vision encoder, which are naturally further apart from text embeddings, while deeper layers are closer to the language modeling supervision and therefore more aligned.

---

> ### Author Response · Authors · 2025-11-25
>
> * Regarding safety, our studies show that a larger modality gap, which is dominated by the early layers, correlates with a higher unsafe rate. This agrees with findings from prior work \[2\], which shows that LLMs largely determine whether an input is ethical or malicious in the early layers, where hidden states for benign versus jailbreak prompts diverge most strongly. A large early-layer modality gap may therefore disrupt these safety-relevant differences and cause the model to misclassify harmful visual inputs. These observations motivated our decision to minimize the modality gap only at the first layer.
>   * Why only regularizing the first layer: As MIR is decreasing for deeper layers, reducing MIR of the first layer automatically results in smaller MIR for deeper layers. Thus, directly regularizing deeper layers may not be necessary but introduces further complexity. Importantly, the strength of regularization should be *layer-dependent*: early layers have a large modality gap and can accommodate a modest pull toward the text space, but higher layers naturally have a much smaller gap and are therefore far more sensitive. Using the same regularizer across multiple layers makes the pull disproportionately strong for these deeper layers, causing their representations to collapse and harming utility. Tuning per-layer regularization would also be very expensive. As shown in Table 5 (also see lines 482-485), regularizing the first three layers yields a higher unsafe rate, as it harms the model performance.
>   * Regularizing the fine-tuning stage. Regularizing the fine-tuning stage has a much more drastic effect on the model’s performance. As discussed in lines 468-469, we observed that regularizing the fine-tuning stage can cause the model to collapse and fail to produce any useful output.
> * 6\. MIR on additional datasets.
>   * We now report MIR on additional datasets such as FigStep and HADES, and the results confirm that the same trend appears across more diverse image distributions. We note that we do not aim to fully close the modality gap as it’d harm the model performance. However, as we showed, slightly lowering the modality gap considerably improves safety without harming the performance.
>
> | MIR-PT | Default | HADES | FigStep | Avg | Var |
> | :---- | ----- | ----- | ----- | ----- | ----- |
> | Sample Size | 500 | 500 | 350 |  |  |
> | ReGAP | 2.14 | 2.55 | 2.75 | 2.48 | 0.06 |
> | LLaVA-7B-LoRA | 3.42 | 3.38 | 3.58 | 3.46 | 0.01 |
>
> | MIR-FT | Default | HADES | FigStep | Avg | Var |
> | :---- | ----- | ----- | ----- | ----- | ----- |
> | Sample Size | 500 | 500 | 350 |  |  |
> | ReGAP | 1.85 | 1.78 | 1.82 | 1.82 | 0.01 |
> | LLaVA-7B-LoRA | 2.82 | 2.75 | 2.84 | 2.8 | 0.01 |
>
> * 7\. Template and typo issues.
>   * Thank you for pointing these out. We have corrected the template year, fixed all typos mentioned by the reviewer, and carefully proofread the full manuscript.
> * 8\. Missing ethics statement.
>   * Thank you for the suggestion\! We have added a dedicated ethics statement to the revised version, in addition to a detailed reproducibility section following the ICLR guidelines.

---

> ### Comment · Reviewer_1y89 · 2025-11-27
>
> I really appreciate the authors' comprehensive response, combined with the new experiments they did. The new experiments on the new datasets show the effectiveness of their method and its potential promises. The utility scores on different benchmarks also seem to suggest that their method does not hurt utility on those benchmarks.
>
> However, my question now is, what are some other utility benchmarks, and in general, other capabilities that are actually hurt by this new pretraining method? This is super important to know because the authors are actually introducing a new pretraining method which, if it's going to be practical and impactful, needs to be on par with the traditional pretraining in the general utilities of the model; otherwise, it's not worth doing pretraining again. In addition, modality gap might not be a bug, but a feature that might be critical to exist for some other capabilities of MLLMs (e.g., spatial reasoning, detailed chart understanding, compositional reasoning, etc).
>
> All in all, I think it's of great importance to better study the effects of this way of doing pretraining on the general capabilities of the model to a much more comprehensive extent; otherwise, the method is not useful given its constraint of requiring us to do pretraining from scratch, given that all good models that exist have been pretrained and instruct-tuned already. More importantly, the authors show that this method does not work on already fine-tuned models, which makes it less practical and less favorable compared to standard safety training. Ultimately, an MLLM that appears “safer” because its modality gap is closed but suffers capability degradation or emergent unwanted behaviors elsewhere is not a meaningful improvement.

---

> > ### Author Response · Authors · 2025-11-27
> >
> > **Practicality.** We agree that all good models that exist have been pretrained and instruct-tuned already. This is exactly why safety improvement during pretraining can have a much larger impact than fine-tuning methods. If model designers incorporate the insights of our method to improve safety during pretraining (by slightly reducing the modality gap), this immediately improves safety for all downstream users on any downstream tasks. Importantly, our pretraining regularization does not require any safety data and thus can address general safety vulnerabilities. In contrast, fine-tuning or inference-time methods require safety data, which is difficult and expensive to collect. Thus, such data is often limited in size and quality, which limits the effectiveness of post-training safety alignment methods. Importantly, post-training methods are task specific and cannot address general vulnerabilities.
> >
> > All in all, both pretraining and post-training methods are required to ensure safety of LVLMs and our pretraining method easily stacks with other safety alignment methods to further boost safety of LVLMs. We believe this is an important step towards safer and more trustworthy AI.
> >
> > \[1\] Liu, Haotian, et al. "Improved baselines with visual instruction tuning." Proceedings of the IEEE/CVF conference on computer vision and pattern recognition. 2024\.
> >
> > \[2\] Chen, Lin, et al. "Sharegpt4v: Improving large multi-modal models with better captions." European Conference on Computer Vision. Cham: Springer Nature Switzerland, 2024\.
> >
> > \[3\] Gao, Jiahui, et al. "Coca: Regaining safety-awareness of multimodal large language models with constitutional calibration." arXiv preprint arXiv:2409.11365 (2024).
> >
> > \[4\] Gou, Yunhao, et al. "Eyes closed, safety on: Protecting multimodal llms via image-to-text transformation." European Conference on Computer Vision. Cham: Springer Nature Switzerland, 2024\.
> >
> >  \[5\] Zong, Yongshuo, et al. "Safety Fine-Tuning at (Almost) No Cost: A Baseline for Vision Large Language Models." Forty-first International Conference on Machine Learning.
> >
> > \[6\] Wang, Han, Gang Wang, and Huan Zhang. "Steering away from harm: An adaptive approach to defending vision language models against jailbreaks." Proceedings of the Computer Vision and Pattern Recognition Conference. 2025\.
> >
> > \[7\] Wang, Yu, et al. "Adashield: Safeguarding multimodal large language models from structure-based attack via adaptive shield prompting." European Conference on Computer Vision. Cham: Springer Nature Switzerland, 2024\.
> >
> > \[8\] Gao, Jiahui, et al. "Coca: Regaining safety-awareness of multimodal large language models with constitutional calibration." arXiv preprint arXiv:2409.11365 (2024).
> >
> > \[9\] Xu, Shicheng, et al. "Cross-modal safety mechanism transfer in large vision-language models." arXiv preprint arXiv:2410.12662 (2024).
> >
> > \[10\] Liu, Qin, et al. "Unraveling and mitigating safety alignment degradation of vision-language models." Findings of the Association for Computational Linguistics: ACL 2025\. 2025\.

---

> ### Author Response · Authors · 2025-11-27
>
> We thank the reviewer for taking the time to read our response and providing further feedback.
>
> We agree that extensive utility evaluation is important. In our response to Reviewer bKys, we had already expanded our evaluation to 13 utility benchmarks, and we now add six more. In total, we have evaluated 19 **utility benchmarks**, covering topics such as **general perception, spatial and temporal reasoning, multi-step inference, table and chart understanding, scientific and mathematical problem-solving, domain knowledge, and detection of visual hallucinations, factual inconsistencies, and grounding failures.** This is **more comprehensive** than the **11 utility benchmarks** used in LVLM foundation model works such as **LLaVA-1.5 \[1\]** and **ShareGPT4V \[2\]**, as well as the **1–4 utility benchmarks** typically evaluated in other LVLM safety-alignment works \[3–7\]. To the best of our knowledge, no prior safety-alignment work conducts such an extensive utility evaluation. In terms of overall utility, ReGAP remains on par with the baseline, exhibiting a **0.6%** increase. As we discuss in the second part of our answer (see below) ReGAP only slightly reduces the modality gap. This results in safety improvement without negatively affecting general model capabilities.
>
> | Model | MMB | MMStar | Seed | TQA | COCO | SQA | CQA | QBench | Hall | **MMVet** |
> | :---- | ----- | ----- | ----- | ----- | ----- | ----- | ----- | ----- | ----- | ----- |
> | LLaVA-7B-LoRA | 64.1 | 35.2 | 66.8 | 24 | 19.2 | 63 | 12.4 | 54.4 | 28 | 27.7 |
> | ReGap | 61.6 | 35.9 | 63.7 | 22.6 | 17.7 | 65.1 | 10.1 | 54 | 28.3 | 27.5 |
> |  |  |  |  |  |  |  |  |  |  |  |
> | Model | **GQA** | **MME** | **VQAv2** | **POPE** | **MathVista** | **TableVQA** | **MMMU** | **InfoVQA** | **RealWorld** | Avg |
> | LLaVA-7B-LoRA | 37.2 | 1600.5 | 79.1 | 85.5 | 23.5 | 10.3 | 36 | 13.2 | 39.1 | 40.8 |
> | ReGap | 39.3 | 1601.0 | 78 | 85 | 25.2 | 14 | 37.3 | 14 | 51.2 | 41.4 |
>
> **Is the modality gap a feature or a bug?** Modality gap is generally expected between vision and language modalities, otherwise vision and language embeddings become indistinguishable and harm the performance. At the same time, (a large) modality gap is known to be the main reason behind the poor performance on VLMs compared to their LLM component. Intuitively, when text is conditioned on an image in a LVLM, the image introduces a shift in the hidden states of the text and pushes it away from the space of original text embeddings. The shifted representations are not fully understandable by the LLM and this results in a lower performance for LVLM compared to its LLM \[10\]. In terms of safety, as discussed in lines 88–95, safety-aligned LLMs suppress harmful intent through specific hidden-state patterns identified in the text during safety alignment. The distributional drift introduced by the image tokens weakens the safety response (i.e. where the safety mechanisms of the LLM normally activate) and leads to higher unsafe rates. This is not the claim of our paper but is observed and discussed in recent prior work (\[8-10\]).
>
> **Our contribution. Our work does not aim to close the modality gap.** In fact, the important **contribution** of our work is to show that (1) *the **amount** of the modality gap can be slightly reduced to considerably improve the safety without harming model performance*. Intuitively a larger modality gap produces larger semantic drift, which in turn reduces the effectiveness of the safety alignment. Furthermore, (2) we demonstrated that *the modality gap is introduced during **pretraining** and reducing the pretraining modality gap directly improves downstream VLM safety after **fine-tuning***. Based on this observation, we propose a method to reduce the amount of pretraining modality gap by mapping image embeddings (slightly) closer to the space of text embeddings to reduce the modality gap and boost safety of the model. As we already showed in our ablation studies in Table 5, the model is much less sensitive to regularizing the pretraining stage compared to the fine-tuning stage. Therefore, our pretraining regularization does not significantly affect the model's general capabilities.

---

> ### Author Response · Authors · 2025-12-03
> **Adding ShareGPT4V result**
>
> **We finished training ShareGPT4V on its full training data.** Similar to our prior experiments, ShareGPT4V also shows a decreased unsafe rate with slightly better utility score, supporting the generality of our method.
>
> | Model | Unsafe Rate (↓) | Utility (↑) |  |  |  |  |  |  |  |  |  |  |  |  |
> | ----- | :---: | :---: | :---: | :---: | :---: | :---: | :---: | ----- | ----- | ----- | ----- | ----- | ----- | ----- |
> |  | HADES Orig | MMB | MMStar | QBench | SEED | SQA | Hall | MMMU | TableVQA | MME | RealWorld | POPE | GQA | Avg |
> | ShareGPT4V | 70.50% | 66.4 | 34.6 | 55.7 | 60.7 | 63.9 | 23 | 30.6 | 12.5 | 1566.61 | 43.9 | 86.6 | 63.2 | 49.8 |
> | ReGAP | 62% | 60 | 32 | 58.4 | 63.2 | 64.2 | 23 | 35.3 | 14.1 | 1613.9 | 48.8 | 85.7 | 59.5 | 50.2 |

---

### Official Review · Reviewer_Y4x6 · 2025-10-30

**Soundness:** 3
**Presentation:** 4
**Contribution:** 2
**Rating:** 4
**Confidence:** 3

**Summary:**

Existing studies reveal that Large Vision-Language Models (LVLMs) suffer from significant safety degradation, blank or irrelevant images can trigger LVLMs to produce harmful responses to prompts that would otherwise be safely refused in text-only contexts. In this study, the authors reveal that the amount of modality gap between text and image embedding is strongly inversed correlated with LVLM safety. Based on this observation, they propose a regularization technique, RECAP, that explicitly reduces the modality gap during pretraining. The experiments showcase that RECAP helps improve the safety characteristics of the LVLM by upgrading a much higher model utility.

**Strengths:**

1.	This paper proposes a novel idea about upgrading the LVLMs’ safety, which is not intuitive but makes their method distinct from other methods.

2.	The generalization capability of the proposed method should be justified. In the experiments, only three LVLMs are tested. I suggest testing more LVLMs to verify the proposed method, like more models in Figure 1.

**Weaknesses:**

1.	Since the proposed method is not intuitive, it is crucial to verify the observation of the correlation between the modality integration rate and unsafe rate. The authors have drawn Figure 1 to verify their motivation, but there is a lack of discussion about the theory behind the observation. Another concern is that only several LVLMs are displayed, and it is not solid to obtain a clear conclusion.

2.	The deep discussion on the existing safety alignment and the proposed methods is missing from the paper. Since it is a different solution to the LVLM's safety issues, the authors should justify their method carefully.

**Questions:**

1.	Is there any theory or larger-scale data to support the correlation between the modality integration rate and unsafe rate?

2.	What are the advantages and disadvantages of the proposed method and existing studies modifying different components of LVLMs?

---

> ### Author Response · Authors · 2025-11-25
>
> We thank the reviewer for recognizing the novelty of and the distinctiveness of our approach.
>
> Weaknesses:
>
> 1. Here is the intuition of our method. As discussed in lines 88–95, safety-aligned LLMs suppress harmful intent through specific hidden-state patterns identified in the text during safety alignment. In LVLMs there is a modality gap between the image and text embeddings, i.e. image embeddings are far away from text embeddings. When text is conditioned on an image in a LVLM, the image introduces a shift in the hidden states of the text and pushes it away from the space of original text embeddings (that are not conditioned on images), where the safety mechanisms of the LLM normally activate. This distributional drift weakens the safety response and leads to higher unsafe rates. This phenomenon is also observed and discussed in prior work (\[1\], \[2\]).
>    The main contribution of our work is to show that the amount of modality gap has a strong negative correlation with safety of an LVLM. Intuitively a larger modality gap produces larger semantic drift, which in turn reduces the effectiveness of the safety alignment. Furthermore, we demonstrated that the modality gap is introduced during pretraining and reducing the pretraining modality gap directly improves downstream VLM safety after fine-tuning. Based on this observation, we propose a method to reduce the amount of pretraining modality gap by mapping image embeddings closer to the space of text embeddings to reduce the amount of shift and boost safety of the model.
>    1. Regarding the number of models in Figure 1, we note that our MIR study have included LVLMs with diverse architectures, sizes, and training pipelines: LLaVA 1.5, Qwen2 VL, Qwen2.5 VL, ShareGPT4V, MiniGPT 4, and InternVL3, as well as LLaVA1.5 \+ RobustCLIP and LLaVA1.5 \+ SimCLIP.
>    2. Additionally, we have included several new models in the MIR study in Fig 1\. The updated correlation is 0.78, and remains a very strong correlation.
>
>
>
> | Model | MIR | Unsafe Rate |
> | :---- | ----: | ----: |
> | DeepSeek-VL-7B | 2.55 | 36.1 |
> | Gemma-4B | 0.87 | 19.2 |
> | SmolVLM2-2.2B | 2.74 | 55.1 |
> | TinyLLaVA-0.5B | 2.16 | 62.6 |
>
>
>
> 2. We have discussed existing categories of methods for improving safety of LVLMs in the second paragraph of our introduction and in more details in our related work section (lines 97-124). We reiterate below:
> 3. Prior work can broadly be divided into three classes: (1) robustifying the vision encoder, (2) safety fine-tuning, and (3) inference-time defenses. Our work complements these techniques by addressing pretraining and can be stacked with them. Existing safety alignment methods intervene at different stages of the LVLM pipeline, but each class carries clear limitations. We show them below:
>
>    1. Training-time safety tuning
>       1. Examples: VLGurad, SPA-VL, Arondight, Red Teaming Visual Language Models
>       2. Pros:
>          1. Directly teaches safety behavior.
>          2. Can leverage high-quality human preference / safety datasets.
>       3. Cons:
>          1. Requires curated safety datasets, which can be very expensive.
>          2. Safety datasets are incomplete, so generalization to unseen harmful images is weak.
>          3. Extra training time is required.
>          4. Overtraining can lead to decreasing downstream model utility or false positive rejection.
>    2. Robust CLIP encoders
>       1. Examples: RobustCLIP, SimCLIP
>       2. Pros:
>          1. Improves safety especially for visually encoded harmful intent.
>       3. Cons:
>          1. Requires additional training of CLIP on large-scale image–caption dataset
>          2. Causes large utility degradation (7-15% degradation)
>    3. Purification-based Inference-time methods
>       1. Examples: Diffusion sanitization, I2T, PSA-VLM
>       2. Pros:
>          1. No need to modify LVLM weights
>          2. Can sanitize harmful visual inputs directly.
>       3. Cons:
>          1. May distort clean images, harming utility.
>          2. Add latency during inference time.
>    4. Auxiliary-Detector / Prompt-Guidance Methods
>       1. Examples: MLLM-Protector, BlueSuffix, ETA, Adashield
>       2. Pros:
>          1. Easy to plug in without retraining LVLM.
>       3. Cons:
>          1. Cause false positives refusals.
>          2. Add latency during inference time.
>          3. Depend entirely on detector accuracy.
>    5. Steering-based methods
>       1. Examples: CMRM, CoCA, TGA
>       2. Pros:
>          1. No additional training cost
>       3. Cons:
>          1.  Learned from limited safe/harmful datasets and does not generalize well to unseen attack types.
>          2. Can over-steer, reducing model utility and output quality.

---

> ### Author Response · Authors · 2025-11-25
>
> Questions:
>
> 1. See Weakness 1
> 2. See Weakness 2
>    \[1\] Gao, Jiahui, et al. "Coca: Regaining safety-awareness of multimodal large language models with constitutional calibration." *arXiv preprint arXiv:2409.11365* (2024).
>    \[2\] Xu, Shicheng, et al. "Cross-modal safety mechanism transfer in large vision-language models." *arXiv preprint arXiv:2410.12662* (2024).
>    \[3\] Liu, Qin, et al. "Unraveling and mitigating safety alignment degradation of vision-language models." *Findings of the Association for Computational Linguistics: ACL 2025*. 2025\.

---

### Official Review · Reviewer_bKys · 2025-11-01

**Soundness:** 3
**Presentation:** 3
**Contribution:** 3
**Rating:** 8
**Confidence:** 4

**Summary:**

This paper addresses the important problem that integrating a vision module into a large language model (LLM) often leads to significant safety degradation. Existing approaches mainly focus on post-hoc safety finetuning or inference-time steering using safety directions. In contrast, this paper offers a fresh perspective by revealing a strong inverse correlation between the safety of vision-language models (VLMs) and the modality gap. The authors propose that this issue originates from what they call the pretraining modality gap. Specifically, since visual features differ substantially from the distribution of text embeddings that the LLM was originally trained on, the model’s inherent safety and robustness mechanisms can be disrupted or bypassed. This observation is closely related to prior works such as Wang et al. (2024c) and Liu et al. (2024a), which explore inference-time interventions and identify the safety direction between visual and textual modalities. The main contribution of this paper lies in tracing the root cause of safety degradation back to the pretraining stage and proposing a new REGAP method to mitigate this issue. Overall, the idea is clear, well-motivated, and convincingly presented.

**Strengths:**

1.	The paper clearly defines the problem and provides empirical evidence that models with larger modality gaps tend to exhibit lower safety levels in VLMs.

2.	It offers a novel finding that the modality gap originating from pretraining is a key factor driving safety degradation in VLMs, and that this gap persists even after fine-tuning.

3.	The proposed REGAP method is simple, efficient, and does not require additional safety data or architectural modifications, making it easy to apply.

**Weaknesses:**

1.	The second paragraph of the introduction could be clarified. Some of the discussed methods do not have the stated limitations (e.g., increased inference time), and others have demonstrated good generalization performance. Clarifying these points early on would strengthen the motivation for the proposed method.

2.	When reducing the distance between image and text token embeddings, a key concern is whether this might cause the two modalities to become overly mixed, potentially harming performance. Therefore, utility tests are particularly important in this work. Including more detailed utility analyses would greatly strengthen the paper’s empirical contribution.

3.	The paper describes REGAP as a lightweight method, but this characterization is somewhat ambiguous. Since the approach involves additional pretraining, it could be more computationally expensive than inference-time intervention methods. It would be beneficial to refine or clarify what is meant by “lightweight” in this context.

**Questions:**

Please check the weaknesses. More utility results will be appreciated.

---

> ### Author Response · Authors · 2025-11-25
>
> We thank the reviewer for their valuable feedback and recognizing the novelty of identifying modality gaps as a root cause of safety degradation, and the simplicity and practicality of ReGAP.
>
> Weaknesses & Questions:
> 1. Clarifications in the introduction.
>    Thank you so much for bringing this inaccuracy to our attention. We have revised the second paragraph of our introduction accordingly. Moreover, we have added Table 6 to the appendix in our revision for more detailed comparisons of related work.
>
> 2. More utility tests
>    1. First, as shown in Fig. 1(a) and Fig. 1(b) (lines 162–170), ReGAP slightly reduces MIR, indicating that it does not over-mix the modalities. Our empirical results further confirm that ReGAP may only slightly lower the performance. We show the full utility analysis below (also in appendix of our original submission, Table 11, line 920-930).
>    2. Second, we evaluated utility on nine datasets: COCO, TextVQA, ChartQA, MMBench, MMStar, ScienceQA, SEEDBench, Q-Bench, and HallusionBench. This coverage is substantially broader than prior work:
>       1. VLGuard: 4 datasets (SQA, VizWiz, MMB, MM-Vet)
>       2. CoCA: 4 datasets (MME, MM-Vet, GQA, VQA)
>       3. ESCO: 3 datasets (MME, MM-Vet, MMB)
>
>
> Here, we provide the full utility analysis and include an additional dataset for evaluation. We see that in terms of utility, ReGAP remains on par with the baseline (with only 0.8% drop). We have included these results into our revision, Table 7\.
>
> | Model | MMB | MMStar | Seed | TQA | COCO | SQA | CQA | QBench | Hall | MMVet | GQA | MME | VQAv2 | Avg |
> | :---- | ----- | ----- | ----- | ----- | ----- | ----- | ----- | ----- | ----- | ----- | ----- | ----- | ----- | ----- |
> | LLaVA-7B-LoRA | 64.1 | 35.2 | 66.8 | 24 | 19.2 | 63 | 12.4 | 54.4 | 28 | 27.7 | 37.2 | 1383 | 79.1 | 44.6 |
> | ReGap | 61.6 | 35.9 | 63.7 | 22.6 | 17.7 | 65.1 | 10.1 | 54 | 28.3 | 27.5 | 39.3 | 1297.8 | 78 | 43.8 |
>
> 3. Unlike safety fine-tuning methods that require an additional training stage and safety data, ReGAP is designed to "replace" the standard LVLM pretraining, not to add an additional stage on top of it. Instead of starting from a pretrained model, we pretrain (with ReGAP regularizer) from an uninitialized model on the original pretraining data, and this regularized pretraining adds only \~5% additional training cost (also shown in Table 1). In this setting, “lightweight” refers to the pretraining cost relative to the original LVLM pretraining pipeline (i.e., 1.05). Compared to other methods that require substantial extra training on top of existing models, such as RobustCLIP, our approach is significantly more efficient. As we have shown, robust vision encoders, or inference-time methods (which require safety data and some processing to e.g. find a steering vector or purify the model’s inputs or outputs) can be easily stacked with ReGAP to further boost safety.

---

> ### Author Response · Authors · 2025-11-27
>
> We further expanded our utility evaluations by adding six additional benchmarks, bringing the total to **19**. These benchmarks cover a wide range of capabilities, including **general perception, spatial and temporal reasoning, multi-step inference, table and chart understanding, scientific and mathematical problem-solving, domain knowledge, and the detection of visual hallucinations, factual inconsistencies, and grounding failures.** This evaluation is more comprehensive than the **11** utility benchmarks used in LVLM foundation model works such as LLaVA-1.5 \[1\] and ShareGPT4V \[2\], as well as the **1–4** utility benchmarks typically adopted in other LVLM safety-alignment works \[3–7\]. To the best of our knowledge, no prior safety-alignment work conducts such an extensive utility evaluation. In terms of overall utility, ReGAP remains on par with the baseline, exhibiting a **0.6%** increase.
>
> | Model | MMB | MMStar | Seed | TQA | COCO | SQA | CQA | QBench | Hall | **MMVet** |
> | :---- | ----- | ----- | ----- | ----- | ----- | ----- | ----- | ----- | ----- | ----- |
> | LLaVA-7B-LoRA | 64.1 | 35.2 | 66.8 | 24 | 19.2 | 63 | 12.4 | 54.4 | 28 | 27.7 |
> | ReGap | 61.6 | 35.9 | 63.7 | 22.6 | 17.7 | 65.1 | 10.1 | 54 | 28.3 | 27.5 |
> |  |  |  |  |  |  |  |  |  |  |  |
> | Model | **GQA** | **MME** | **VQAv2** | **POPE** | **MathVista** | **TableVQA** | **MMMU** | **InfoVQA** | **RealWorld** | Avg |
> | LLaVA-7B-LoRA | 37.2 | 1600.5 | 79.1 | 85.5 | 23.5 | 10.3 | 36 | 13.2 | 39.1 | 40.8 |
> | ReGap | 39.3 | 1601.0 | 78 | 85 | 25.2 | 14 | 37.3 | 14 | 51.2 | 41.4 |
>
> \[1\] Liu, Haotian, et al. "Improved baselines with visual instruction tuning." Proceedings of the IEEE/CVF conference on computer vision and pattern recognition. 2024\.
>
> \[2\] Chen, Lin, et al. "Sharegpt4v: Improving large multi-modal models with better captions." European Conference on Computer Vision. Cham: Springer Nature Switzerland, 2024\.
>
> \[3\] Gao, Jiahui, et al. "Coca: Regaining safety-awareness of multimodal large language models with constitutional calibration." arXiv preprint arXiv:2409.11365 (2024).
>
> \[4\] Gou, Yunhao, et al. "Eyes closed, safety on: Protecting multimodal llms via image-to-text transformation." European Conference on Computer Vision. Cham: Springer Nature Switzerland, 2024\.
>
>  \[5\] Zong, Yongshuo, et al. "Safety Fine-Tuning at (Almost) No Cost: A Baseline for Vision Large Language Models." Forty-first International Conference on Machine Learning.
>
> \[6\] Wang, Han, Gang Wang, and Huan Zhang. "Steering away from harm: An adaptive approach to defending vision language models against jailbreaks." Proceedings of the Computer Vision and Pattern Recognition Conference. 2025\.
>
> \[7\] Wang, Yu, et al. "Adashield: Safeguarding multimodal large language models from structure-based attack via adaptive shield prompting." European Conference on Computer Vision. Cham: Springer Nature Switzerland, 2024\.

---

### Official Review · Reviewer_tbDC · 2025-11-01

**Soundness:** 3
**Presentation:** 2
**Contribution:** 2
**Rating:** 6
**Confidence:** 4

**Summary:**

This paper addresses a critical problem: the significant safety degradation observed in Large Vision-Language Models (LVLMs) compared to their base Large Language Models (LLMs). The authors identify a "modality gap" between image and text embeddings and demonstrate a strong inverse correlation between this gap and the safety of LVLMs. They propose REGAP, a lightweight regularization method applied during the pretraining stage. REGAP explicitly reduces this modality gap using a computationally efficient surrogate loss without requiring additional safety data.

**Strengths:**

1. the paper makes a novel observation of how the amount of modality gap is related with LVLMs safety.
2.The problem and the proposed method are clearly illustrated.
3.The proposed REGAP requires no safety data and is complementary to other defense methods, making it practical.

**Weaknesses:**

1. There is a certain risk of exaggeration in claims about data/model coverage and generality:  Experiments are validated on LLaVA / ShareGPT4V / MiniGPT-4, etc., but ShareGPT4V uses only one-quarter of the data and MiniGPT-4 uses a subset (Sec.6.1); moreover, if the effects under different vision encoders (other than CLIP-ViT-L) or larger LLMs (e.g., 13B/70B) have not been fully verified, the assertion of generality appears overly broad.
2. Dependence on LLM-judge (Beaver-dam-7B): Automated judges can be biased or have failure modes. It is suggested to (a) report human validation on random subset; (b) evaluate sensitivity to different judge models/thresholds.

**Questions:**

1. It is mentioned in Section 4 (lines 245–246) of sampling 100 image–prompt pairs to compute MIR which might not be sufficient considering the randomness. Could you report the diversity of the samples or the variance across different sample sizes (e.g., 50/100/500/1000) to justify that 100 samples are representative of the overall dataset.
2. Some experimental details are lacked: The paper states that α is calculated and fixed during the warm-up period (Sec.5), but maybe it lacks key pretraining configurations such as warm-up duration, learning rate, batch size, sampling seed, and the sensitivity of REGAP to data of different scales.
3. The generality regarding fusion architectures is in question: The experiments primarily focus on LLaVA-style architectures (MLP projector) and MiniGPT-4 (Q-Former) (lines 740-754). How would REGAP perform on models with different vision-language fusion mechanisms, such as those employing cross-attention (e.g., Flamingo, BLIP-2)? The paper generalizes across datasets (ShareGPT4V) and a different projector (MiniGPT-4), but its applicability to fundamentally different fusion architectures is not demonstrated.

---

> ### Author Response · Authors · 2025-11-25
>
> We thank the reviewer for their valuable feedback and recognizing the novelty of our observation, the clarity of our method, and the practicality of ReGAP.
>
> Weaknesses:
>
> * 1\. Claims about coverage and generality.
>   * To show the validity of our claims, we ran new experiments to train MiniGPT4 with ReGAP on its full dataset. The following table shows that ReGAP effectively  decreases the unsafe rate of MiniGPT4 and yields a slightly better (+0.5%) utility score than the original model, supporting the generality of our method. We have added this into our revision (Table 3\)
>
>
> | Model | Unsafe Rate (↓) | Utility (↑) |  |  |  |  |  |  |  |
> | ----- | :---: | :---: | :---: | :---: | :---: | :---: | :---: | :---: | :---: |
> |  | HADES Orig | MMB | MMStar | QBench | GQA | SEED | SQA | Hall | Avg |
> | MiniGPT4 | 68 | 49.5 | 30.7 | 41.7 | 27.9 | 38.6 | 51.8 | 24.1 | 37.8 |
> | \+ReGAP | 59.8 | 49.3 | 28.7 | 40.5 | 24.2 | 40.3 | 55 | 29.8 | 38.3 |
>
>
>
>   * For ShareGPT4, full-dataset training requires roughly 384 GPU-hours on NVIDIA A40. We are still running the experiment and will post the results if we get them in time. Otherwise, we’ll add it to our final version.
>   * Regarding different vision encoders or larger LLMs, we note that ReGAP is applied at the LLM input embedding layer by penalizing the L2 distances between vision tokens (output of vision encoder) and their corresponding projected text tokens (LLM input). I.e. ReGAP only regularizes the projector/mapping (to map vision tokens closer to the text space) and does not change the output of the vision encoder. Therefore, we do expect that our conclusions are not dependent on the choice of vision and language encoders. Due to computational constraints, we are not able to train models with very large LLMs (13B/70B).
>     To make our statements more precise, we have limited the scope of our claims to open models of size up to 7B in the introduction of our revision (with blue font).
> * 2\.Human evaluation and different judge models/thresholds
>   * In our revised version, we have incorporated two additional evaluations: (a) human validation on a randomly sampled subset of 100 generated responses, and (b) robustness checks across multiple judge models like GPT-4o and scoring thresholds (in our original submission we reported the default threshold \= 0.5 following HADES \[1\]). We report our new results below:
>
> | Model | t=0.4 | t=0.45 | t=0.5(d) | t=0.55 | t=0.6 | t=0.7 | GPT-4o | Human Judge Score (100) |
> | :---: | :---: | :---: | :---: | :---: | :---: | :---: | :---: | :---: |
> | LLaVA-7B-LoRA (Baseline) | 74.40% | 70.27% | 68.10% | 62.80% | 58.00% | 48.00% | 82.90% | 86% |
> | ReGAP | 69.30% | 66.13% | 63.20% | 60.50% | 56.40% | 46.53% | 73.60% | 73% |
> | ReGAP \+ RobustCLIP | 50.40% | 47.60% | 44.80% | 42.40% | 40.27% | 33.10% | 55.70% | 59% |
>
> We observe that ReGAP consistently reduces unsafe responses across all judge models, including human judge, and thresholds.
>
> Question:
> * 1\. MIR sample size questions
>   * Per reviewer’s suggestion, we computed MIR across multiple sample sizes (50, 100, 500\) and different datasets (FigStep, HADES). Below, we report the variance of the MIR estimates. We observe that MIR remains stable across sample sizes, and ReGAP consistently achieves lower MIR regardless of sample size or dataset.
>   * Note that, for FigStep, the dataset contains only 350 images, so we report results up to this limit. For the other datasets (e.g., HADES), computing MIR with 1000 samples is computationally expensive, so here we report for up to 500 samples.
>
> | MIR-PT | Default |  |  | HADES |  |  | FigStep |  |  | Avg | Var |
> | :---: | :---: | :---: | :---: | :---: | :---: | :---: | :---: | :---: | :---: | ----- | ----- |
> | Sample Size | 50 | 100 | 500 | 50 | 100 | 500 | 50 | 100 | 350 |  |  |
> | ReGAP | 2.13 | 2.18 | 2.14 | 2.62 | 2.63 | 2.55 | 2.78 | 2.80 | 2.75 | 2.51 | 0.08 |
> | LLaVA-7B-LoRA | 3.42 | 3.43 | 3.42 | 3.39 | 3.38 | 3.38 | 3.58 | 3.56 | 3.58 | 3.46 | 0.01 |
>
> | MIR-FT | Default |  |  | HADES |  |  | FigStep |  |  | Avg | Var |
> | :---: | :---: | :---: | :---: | :---: | :---: | :---: | :---: | :---: | :---: | ----- | ----- |
> | Sample Size | 50 | 100 | 500 | 50 | 100 | 500 | 50 | 100 | 350 |  |  |
> | ReGAP | 1.84 | 1.85 | 1.85 | 1.78 | 1.78 | 1.78 | 1.82 | 1.82 | 1.82 | 1.82 | 0.01 |
> | LLaVA-7B-LoRA | 2.82 | 2.82 | 2.82 | 2.75 | 2.74 | 2.75 | 2.83 | 2.83 | 2.84 | 2.80 | 0.01 |

---

> ### Author Response · Authors · 2025-11-25
>
> * 2\.Experimental details (warm-up duration, LR, sensitivity, etc.)
>   * Our experimentation details can be found in Table 15 and Table 16 in Appendix of our original submission (line 1188 \- line 1227). We also reiterate our pretraining details here:
>   * As for the sensitivity of ReGAP to data of different scales, we note that we have trained ReGAP across widely varying pretraining sizes: MiniGPT4  (full data with 5M image–text pairs in our new experiments, and a subset of 1.25M-pairs in our original submission), ShareGPT-4V (a subset of 0.31M-pairs), and LLaVA-1.5 (558K pairs). ReGAP consistently improves safety alignment in all settings.
>
> | Epoch | Batch Size/GPU | Grad Accu | LR | WD | Warmup Ratio | LR Scheduler | TF32 | Max Seq Length | Grad Checkpoint | Sampling Seed |
> | :---- | :---- | :---- | :---- | :---- | :---- | :---- | :---- | :---- | :---- | :---- |
> | 1 | 32 | 1 | 1×10⁻³ | 0 | 0.03 | Cosine | True | 2048 | True | 42 |
>
> * 3\. Generality regarding fusion architectures
>   * First, we note that MiniGPT4 is a BLIP-2–style model and shares most of the same architectural components, including the CLIP image encoder and the Q-Former–based connector. This means our experiments already include a representative BLIP-2 family architecture, and thus our method naturally extends to BLIP-2 training.
>   * Moreover, ReGAP regularizes the connector output, and the regularization is not tied to any specific projector structure. Regardless of whether the connector is an MLP projector (LLaVA-style), a Q-Former (BLIP-2/MiniGPT-4), or a perceiver-resampler with cross-attention (Flamingo-style), all architectures would output a sequence of visual tokens to input to the LLM. Because ReGAP regularizes these projected tokens directly, it remains applicable across different fusion mechanisms.
>
> [1] Li, Yifan, et al. "Images are achilles’ heel of alignment: Exploiting visual vulnerabilities for jailbreaking multimodal large language models." European Conference on Computer Vision. Cham: Springer Nature Switzerland, 2024.

---

> ### Author Response · Authors · 2025-12-03
> **Adding ShareGPT4V Result**
>
> **We finished training ShareGPT4V on its full training data.** Similar to our prior experiments, ShareGPT4V also shows a decreased unsafe rate with slightly better utility score, supporting the generality of our method.
>
> | Model | Unsafe Rate (↓) | Utility (↑) |  |  |  |  |  |  |  |  |  |  |  |  |
> | ----- | :---: | :---: | :---: | :---: | :---: | :---: | :---: | ----- | ----- | ----- | ----- | ----- | ----- | ----- |
> |  | HADES Orig | MMB | MMStar | QBench | SEED | SQA | Hall | MMMU | TableVQA | MME | RealWorld | POPE | GQA | Avg |
> | ShareGPT4V | 70.50% | 66.4 | 34.6 | 55.7 | 60.7 | 63.9 | 23 | 30.6 | 12.5 | 1566.61 | 43.9 | 86.6 | 63.2 | 49.8 |
> | ReGAP | 62% | 60 | 32 | 58.4 | 63.2 | 64.2 | 23 | 35.3 | 14.1 | 1613.9 | 48.8 | 85.7 | 59.5 | 50.2 |

---

### Author Response · Authors · 2025-12-03
**Summary to AC**

**Summary to AC**

Dear AC,

Thank you so much for your extra effort in reviewing our paper and rebuttal. Below, we summarize our contributions and provide a brief summary of the key changes we made to our revision.

**Contributions:** By studying 14 open-weight LVLMs, for the first time, we:

* Showed that the magnitude of the modality gap (distance between image and text embeddings) is strongly and consistently inversely correlated with LVLM safety.
 * Identified that the modality gap emerges during pretraining and persists throughout fine-tuning. This identifies the pretraining-stage modality gap as a previously overlooked root cause of safety degradation in LVLMs.
 * Propose ReGAP, a lightweight pretraining-stage regularizer that explicitly reduces the modality gap during pretraining. ReGAP:
    * improves LVLM safety by up to 16.3%,
    * preserves or slightly improves utility,
    * is applicable across **different architectures and training data distributions**,
    * requires no safety data,
    * incurs minimal (\<5%) computational overhead, and
    * can stack with existing post-training defense methods to achieve up to 18.2% additional safety gains.
  * The novelty of our findings, the simplicity and effectiveness of ReGAP, and our extensive experiments and well-written manuscript is acknowledged by all the reviewers.

**Changes to our revision (with blue font):**

* ***Added 10 new utility benchmarks, totaling 19 (Table 7),*** which shows strongly that ReGAP “preserves model utility performance”, while considerably boosting its safety.
* Expanded our correlation analysis to additional LVLMs (DeepSeek-VL, Gemma-4B, SmolVLM2, TinyLLaVA) in Table 1, totaling **14 models**, and confirmed the consistently strong correlation between MIR and unsafe rate (**0.78**, Table 1).
* Added **results** for training MiniGPT-4 and ShareGPT4V on their **full training data** (Table 3 and 4), all showing strong unsafe-rate reductions and on-par utility relative to standard training. These results demonstrate the applicability and strong performance of ReGAP across **different model sizes, connector architectures, and dataset distributions**.
* Included **human evaluations**, **GPT-4 judge evaluations**, and **unsafe-rate curves across thresholds 0.4–0.7**, all of which consistently show that ReGAP reduces unsafe behavior under diverse evaluation settings (Table 17).
* Reported full per-layer MIR results for both pretraining and fine-tuning in Table 8 and Table 9, demonstrating that the modality gap is significantly larger in early layers and motivating why ReGAP is applied only to input layers.
* Added Table 6 for detailed comparison of existing safety methods, an Ethics Statement and a Reproducibility Statement, and corrected minor typos.

Thank you again for your time and effort,

Authors

---

### Meta-Review · Area_Chair_Kb8d · 2026-01-07

**Summary:**

This submission proposes a new method to reduce the modality gap for LVLM. The paper received reviews from four reviewers, resulting in divergent scores. Some concerns are addressed during the rebuttal phase, however, there are still some key questions unsolved.

1. The core insight that the modality gap is linked to safety degradation was viewed as already established by prior work.
2. The proposed approach is computationally prohibitive for the vast majority of the community, who rely on fine-tuning already-pretrained models where the method doe not work in this scenario.
3. The paper presents a method during pre-training with extensive validation only on smaller models (up to 7B).

Therefore, AC is to recommend reject at this stage.

**Reviewer Concerns:**

Addressed
1. Reviewer tbDC: the diversity of the samples or the variance across different sample sizes.
2. Reviewer bKys: concerns about Utility are addressed with 19 benchmarks.
3. Reviewer 1y89: Missing ethics statement.

Unresolved
1. Reviewer tbDC: The effects on larger LLMs (13B/70B) have not been verified.
2. Reviewer tbDC: The effects under different vision encoders (other than CLIP-ViT-L) were not verified.
3. Reviewer Y4x6: lack of discussion about the theory behind the observation.
4. Reviewer 1y89: modality gap might not be a bug, but a feature that might be critical to exist for some other capabilities of MLLMs.

**Reviewer Scores:**

Reviewer tbDC might have the score maintained because he/she has already scored an initial positive rate and some issued are not been addressed.
Reviewer bKys might have the score maintained because he/she has already socred an initial positive rate.
Reviewer Y4x6 might have the score maintained since some issued are not been addressed.
Reviewer 1y89 might have the score maintained since his/her main concern has not been solved.

---

### Decision · Program_Chairs · 2026-01-26

Reject